# Analysis on urban scaling characteristics of China's relatively developed cities

**Xingchao Liu** *, **Zhihong Zou**

School of Economics and Management, Beihang University, Beijing, China

* liuxingchao1408@163.com

## Abstract

China is undergoing rapid urbanization, but the speed and stage of urban development are quite heterogeneous among different regions and city types. Understanding the urban scaling characteristics of China's relatively developed cities is important for addressing environmental and social challenges. Within the scope of 114 third-tier-and-above Chinese cities, the research calculate the scaling parameters of various urban development variables with respect to urban population and urban GRP in different city types based on urban scaling quantitative models. Also, univariate and multivariate regression analyses were performed on the factors affecting urban electricity consumption. The research results show that the urban scaling characteristics of Chinese cities differ between different types of cities, industrial cities show unique scaling features compared to commercial cities and mixed-economy cities. Additionally, urban electricity consumption is found to be closely related to urban population, urban construction land area and street lamp number. The results can help different types of cities make targeted policies and provide insights for reducing resource consumption during the urbanization process.

**Data Availability Statement:** All relevant data are available from the China Economic and Social Development Statistical Database of the China National Knowledge Infrastructure (CNKI). Data were retrieved from the China City Statistical Yearbook (http://data.cnki.net/Yearbook/Single/

## 1. Introduction

At present, China's urbanization is rapidly progressing. By 2018, China's urbanization rate had reached 59.58% (From the National Bureau of Statistics). The sizes and numbers of Chinese cities are both growing rapidly [1]. Although China's urbanization rate is quite fast in the world, its urbanization process still lags behind other countries [2, 3]. Besides, the complex Chinese national conditions give China's urbanization unique characteristics [4]. Due to this excessive urbanization speed, Chinese cities' industrial structure, resource allocation, and technological progress do not match with their degree of development [5]. Mastering the process of urbanization in China needs quantitative models in the urban scaling study area.

City is the principal place of human life, and people have always maintained great interest in the development of urban systems [6]. However, due to the existence of various ever-changing systems such as society, economy, and infrastructure, cities can seem very complicated on the surface [7]. The explosive growth and rapid expansion of urban systems have led to fierce

N2020050229) and the China City Construction Statistical Yearbook (http://data.cnki.net/yearbook/Single/N2019060082). The authors of this study had no special access privileges in accessing the data sets which other interested researchers would not have.

**Funding:** The author(s) received no specific funding for this work.

**Competing interests:** The authors have declared that no competing interests exist.

competition for space and resources between different urban systems, so the sizes and shapes of cities follow specific rules [8–10]. In fact, city systems correspond with life characteristics of biological systems [11, 12]. In biological systems, there is a sub-linear power-law relationship between the quality of mammalian species and the metabolic rate of organisms [13, 14]. This allometric growth scale law is modeled as a common feature observed by all biological systems [15]. The similarity between biological systems and urban systems makes it possible to conclude universal applicable urban scaling laws based on the general allometric growth scale models in biology [16].

By following the growth model of biological system, Bettencourt and his colleagues modeled the general scaling of urban systems: most characteristics of urban wealth creation and material energy use show index extensions with the increase of urban population and population interactions [16, 17]. In different geographies or different city scales, the scaling parameters of homogeneous indicators remain consistent [18, 19].

The urban indicators depicting the development of urban systems can be classified into three categories: innovative wealth indicators related to social wealth and social nature, such as inventions, crime rates and so on; physical energy indicators related to individual needs, such as water consumption, electricity consumption and so on [3, 20]; urban infrastructure indicators, such as the number of street lights, the length of the water supply pipeline, and so on [21]. Different categories of urban indicators present different characteristics as a city expands [22, 23].

A city's innovative wealth indicators tend to exhibit super-linear growth proportionality with city expansion [24]. Cities promote urban economic growth, wealth creation, and new ideas by attracting creative and innovative individuals [22, 25]. With the growth of urban talents and innovation, a city's socio-economic performance will exceed the proportional growth of the urban population [26]. The per capita invention and creativity of larger cities are significantly higher than those of smaller cities, and the gap is further increasing, which indicates that a city's innovative inventions have super-linear proportional relationships with the population growth [27–30].

The material and energy indicators of cities tend to be linearly proportional to the expansion of cities due to the close correlations with individual needs [31]. Scholars such as Kennedy explored the material and energy flows of 27 megacities with a population of more than 10 million to verify the consistency between the laws of resource flows in megacities and the general laws of urban scaling [32]. Further, the material and energy flow research on Chinese cities provides supporting evidence for the linear relationships between material energy indicators and city scaling [33].

Urban infrastructure indicators tend to obey sub-linear scaling laws as cities expand, that is, as the city population grows, the physical network usually grows more slowly than the city's scale growth [34, 35]. This is mainly because of the existence of economies of scale [36].

Among the studies of urban scaling laws, how to determine the geographical extent of cities has always been a focus of discussion [37]. The criteria used to classify cities makes a big difference in the effectiveness of urban scaling models [6, 9]. Urban scale parameters are sensitive to urban partition and population size in the process of urban scaling [35, 38]. Research using public census data will continue to dominate the mainstream [39].

China's urban development is hugely unbalanced. Cities of different development levels in different regions show disordered states for both geographical and policy reasons. Based on numerous previous studies, the urban scaling laws can be more obvious in more developed cities. This study focuses on relatively developed cities in China, i.e., cities of the third tier and above on a five-tier scale. These cities have urban administrative units that are subject to high levels of urbanization and are thus more likely to belong to the same "urban system" [7].

With the development of the urban economy, the importance of primary industry will generally decline, while the proportion of secondary and tertiary industries will rise rapidly [40, 41]. The differences between the proportions of the first, second and third industries in different cities could lead to different scaling characteristics; thus, exploring the differences in scaling laws between different types of cities is taken into account in our work, while previous studies have ignored the impact of city type.

In term of variable selection, we have included many more indicators. As independent variables, both urban population and urban GRP are used to describe the scaling characteristics of cities. A broader variety of indicators concerning sustainable urban development are also included as response variables and are analyzed at finer levels.

China's urbanization process consumes a lot of energy, and electricity is an essential component [33, 42]. Electricity is not only the necessary energy directly needed in the commercial and industrial development of cities, but also urban residents' most vital energy in daily life [43, 44]. Most importantly, electricity consumption is one of the primary sources of $CO_2$ emissions [45]. To analyze the electricity consumption during urban scaling, univariate and multivariate regression analysis were conducted on the factors affecting urban electricity consumption in different types of cities. According to the analysis results, policies are recommended.

The main objectives of this research include the following aspects:

Calculation of the scaling parameters of urban development indicators as a function of urban population and urban GRP within the scope of 114 Chinese third-tier-and-above cities, and analysis of whether the scaling characteristics of different types of indicators are consistent with Bettencourt's conclusions;

Exploring the differences in urban scaling laws between industrial cities, commercialized cities and mixed-economy cities, and analyzing the reasons for the differences;

Carrying out univariate and multivariate regression analysis on the factors affecting urban electricity consumption of different types of cities and providing some suggestions according to the research results.

## 2. Materials and methods

### 2.1 Research data

**2.1.1 Data sources.**   The research data mainly come from the China Urban Statistical Yearbook—2017 issued by the Department of Urban Social and Economic Investigation and the China Urban Construction Statistical Yearbook—2016 issued by the Ministry of Housing and Urban-Rural Development of the People's Republic of China [46, 47].

The data from the two yearbooks was cross-checked for data revision, and the China economic and social development statistical database was searched for the remaining missing data [48]. The processed data table containing 51 development variables of 263 Chinese cities was assembled as the original research data.

**2.1.2. Selection of research cities and description of variables.**   As the development of China at this stage is unbalanced and insufficient, the development status varies significantly from city to city. As the level of urban development becomes higher, the laws followed by urban development are more pronounced. Therefore, selecting Chinese cities with better development could help to improve the pertinence of the research.

"2016 China Business Charm Ranking" was published by the "New First tier City Research Institute", a data news project of China Business Weekly, which ranked 338 Chinese prefecture-level cities on five dimensions of plasticity, including the concentration of business resources, urban hubs, urban people's activity, lifestyle diversity and future. According to the

ranking results, there are 4 first tier cities, 15 new first tier cities, 30 second tier cities, 70 third tier cities, 90 fourth tier cities and 129 fifth tier cities. Based on the original data table and city classification results of the ranking, 114 cities, including 4 first tier cities, 15 new first tier cities, 27 second tier cities and 68 third tier cities, were selected for the research. The 114 selected cities are all third-tier-and-above cities. Their urban development is relatively mature and the construction of urban infrastructure is relatively better. The urban population, urban GRP and urban development resource consumption of the 114 cities account for the vast majority of Chinese cities. Exploring the scaling laws of these cities could help in understanding the overall urban development pace in China.

The 114 cities were divided into three different types of cities by the classification criteria proposed by Nelson [49]. Specifically, cities in which the proportion of secondary industry GRP is higher than the national average (the average of 263 cities) plus one standard deviation (58.00%) were classified as industrial cities; cities in which the proportion of tertiary GRP is higher than the average level plus one standard deviation (57.15%) were classified as commercial cities, and the rest were classified as mixed-economy cities. The classification results gave 14 industrial cities, 27 commercial cities and 73 mixed-economy cities in a total of 114 cities.

The 25 variables related to urban scaling selected in the research include urban population, GRP, total urban gas supply, total urban water supply, total urban electricity consumption, and so on. The administrative areas of the selected variables are municipal districts. The municipal district usually has high-level urbanization, massive population density and higher urban GRP. The municipal districts in China best fit the definition of cities in similar research studies.

## 2.2 Urban scaling model

The calculation model commonly used in urban scaling research is the quantitative model proposed by Bettencourt and his colleagues:

$$Y(t) = Y(0) \cdot N(t)^{\beta} \tag{1}$$

N(t) represents a measure of the size of the urban population at time t; Y(0) is a normalized constant; Y(t) can represent a measure of material resources or social activity (e.g., wealth, patents and water consumption); the index β represents the general scaling parameter of urban development indicators with respect to population size. The model applies to cities in different years and different regions.

The leading urban development indicators were divided into infrastructure categories, individual demand categories and innovative wealth categories. The three types of urban indicators showed different scaling characteristics in the process of urban expansion. Between the different types of urban scaling indicators as the population size expands, the main differences are the general scaling parameter β values: β≈1 usually correspond to individual demands; β≈1.1–1.5>1 is usually related to social innovation wealth; β≈0.85<1 is usually associated with urban infrastructure. The differences in β values indicate different categories of indicators and show different scaling ratios as the urban population changes.

As a direct variable reflecting the degree of urban economic development, urban GRP plays a vital role similar to that of the urban population in the expansion of cities; thus, GRP was introduced into the model as a reactive indicator. Similar to model (1), the quantitative model of urban development indicators as a function of urban GRP can be described as:

$$Y(t) = Y(0) \cdot (GRP)^{\beta} \tag{2}$$

where Y(t) indicates the material resource or social activities; Y(0) is the normalization

constant; the general scaling parameter β represents how different urban development indicators vary with GRP.

In order to facilitate the calculation, formula (2) is generally paired in the actual calculation process with:

$$\ln(y) = c + b \cdot ln(x) \tag{3}$$

where y represents the urban development indicators that need to be explained, such as urban electricity consumption; x represents the urban scaling variables used to explain y, and in the model of this study x is urban population and urban GRP; c is the normalization constant; b represents the general scaling parameters of urban development indicators as a function of urban population or urban GRP.

The exploration of urban electricity consumption mainly uses multivariate regression analysis, and its calculation formula can be expressed as:

$$\ln(electricity) = c + \sum_{i=1}^{n} b_i \ln(indicator_i) \tag{4}$$

where electricity represents the city's electricity consumption, which includes the city's total electricity consumption, urban industrial electricity consumption and urban residents' electricity consumption; c is the normalization constant; $indicator_i$ represents the variables affecting electricity consumption; $b_i$ serve as the parameters of explanatory variables of urban electricity consumption.

## 3. Results and discussion

The urban scaling parameters of various urban development indicators were calculated using the urban scaling models and the 2016 urban development yearbook data. A few urban development indicators that show better fitting effects are shown in our results. In addition, univariate and multivariate regression analyses were conducted on the factors affecting electricity consumption of different types of cities. Based on the results, some suggestions are made for reducing urban electricity consumption.

### 3.1 On urban scaling laws

Section 3.1.1 explores the overall scaling characteristics of China's third-tier-and-above cities and section 3.1.2 analyses the differences in the scaling laws between three different types of cities.

**3.1.1 Scaling laws of all third-tier-and-above cities.** Fig 1 shows the unitary regression results of urban GRP and urban population on a logarithmic scale.

As can be seen from Fig 1, for China's 114 third-tier-and-above cities, urban GRP shows a super-linear scaling relationship with the urban population (b = 1.111, $R^2$ = 0.752), which is mainly due to the bidirectional positive feedback between the two. Cities with higher GRP are more mature and have more employment opportunities, thus attracting more urban population. More urban population could promote the further increase of urban GRP. As a result, urban GRP expands in a super-linear manner with urban population increase.

Table 1 shows the scaling parameters of different urban development variables relative to urban population and urban GRP.

In Table 1, it can be seen that urban development indicators related to individual needs, including total urban water supply, total urban electricity supply, fixed asset investment, built-up area and drainage pipeline length, scale linearly with the urban population.

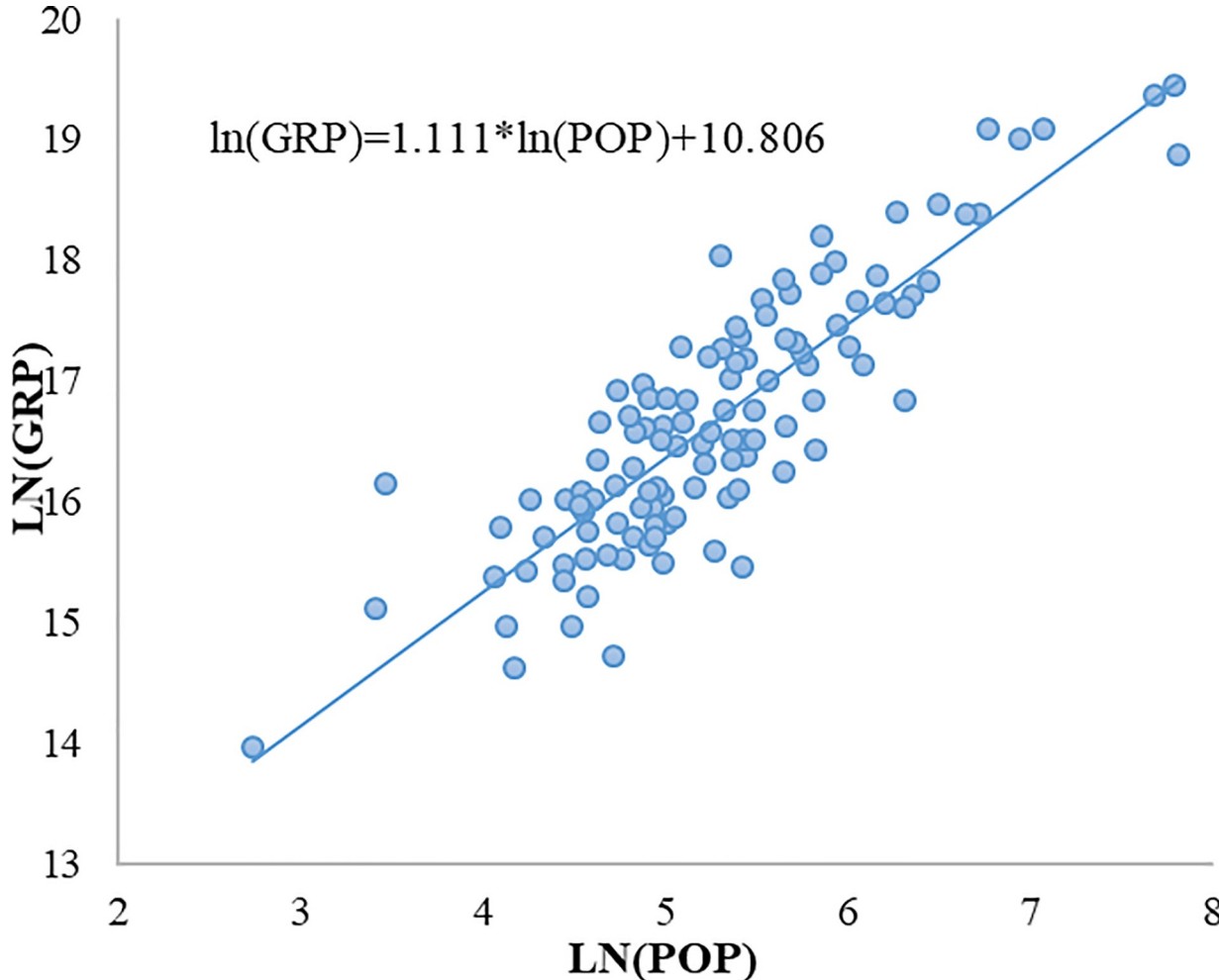

**Fig 1. Logarithmic regression of urban population and urban GRP.** (A) The blue bubble corresponds to the logarithmic population and GRP of each city; (B) The blue slash represents the logarithmic regression line of urban population and urban GRP.

Urban development indicators related to urban infrastructure including road area, park area and street light number expand sub-linearly with the urban population, mainly due to the existence of economies of scale. The construction and operation of the same infrastructure at higher density are more efficient, more economically viable and often result in higher quality services and solutions that are not possible in smaller locations, therefore often leading to economies of scale, which in turn leads to slower urban infrastructure construction speed. It's worth noting that green coverage area expands linearly with population. Green coverage area includes not only park green area, but also residential green area and transportation green area, thus green coverage area is closely related to individual needs which leads to the linear relationship with population.

In general, the scaling parameter values of different types of urban development indicators are consistent with the conclusions that Bettencourt and colleagues have presented.

In Table 1, indicators related to urban energy consumption, including urban total gas supply, urban total water supply, and urban total electricity supply, scale linearly with urban GRP. Other indicators including fixed asset investment, construction land area, drainage pipeline length, road area, park area and street light number show sub-linear scaling with urban GRP,

Table 1. Scaling parameters of urban variables with urban population and urban GRP.

| LN(RV) | With respect to LN(POP): First, Second and Third tier cities (n = 114) | | | With respect to LN(GRP): First, Second and Third tier cities (n = 114) | | |
|---|---|---|---|---|---|---|
| | b | Linearity | Adj-$R^2$ | b | Linearity | Adj-$R^2$ |
| TGS | 1.247 | Super-L | 0.508 | 1.004 | L | 0.552 |
| TWS | 1.047 | L | 0.709 | 0.894 | L | 0.848 |
| WSEC | 0.994 | L | 0.669 | 0.853 | L | 0.81 |
| FAI | 0.947 | L | 0.667 | 0.807 | Sub-L | 0.794 |
| CLA | 0.856 | L | 0.748 | 0.717 | Sub-L | 0.853 |
| DPL | 0.916 | L | 0.635 | 0.784 | Sub-L | 0.765 |
| RA | 0.868 | Sub-L | 0.68 | 0.737 | Sub-L | 0.803 |
| PA | 0.788 | Sub-L | 0.503 | 0.669 | Sub-L | 0.594 |
| GCA | 0.932 | L | 0.637 | 0.816 | Sub-L | 0.802 |
| SLN | 0.761 | Sub-L | 0.548 | 0.675 | Sub-L | 0.707 |
| LN(GDP)~LN(POP):b = 1.111, Adjusted $R^2$ = 0.752 | | | | | | |

**RV:** Response variable

**TGS:** Total gas supply; **TWS:** Total water supply; **WSEC:** Whole society electricity consumption; **FAI:** Fixed asset investment; **CLA:** Construction land area; **DPL:** Drainage pipe length; **RA:** Road area; **PA:** Park area; **GCA:** Green coverage area; **SLN:** Street lamp number.

**Super-L:** Super-Linear; **L:** Linear; **Sub-L:** Sub-Linear; **Adj-$R^2$**: Adjusted $R^2$.

and these indicators can be collectively referred to as urban construction indicators. Urban GRP development is accompanied by energy consumption, while the restriction of material energy use efficiency makes the urban energy consumption follow certain linear proportional relationships with GRP increases. Also, the $R^2$ values of the fitting equations between urban GRP and the indicators are significantly larger than those of the fitting equations between urban population and the indicators. This indicates that compared with the urban population, urban development indicators show stronger correlations with urban GRP, which means Chinese urban scaling characteristics could be better measured by urban GRP than the urban population.

It is worth noting that in urban material energy indicators, the total urban gas supply shows significantly different scaling characteristics comparing to water supply and electricity supply. The total gas supply scales super-linearly with the urban population (b = 1.247), while the total urban water supply and the urban electricity supply show linear proportional characteristics with urban population changes (b = 1.047, b = 0.994).

In Table 1, the correlation between total urban gas supply and urban population is significantly weaker than that between the urban population and the total urban water supply or the total urban electricity consumption. In cities, the use of urban gas supply is applicable mainly for residential households. Urban gas supply is not a necessary choice for residents because urban households have more options for cooking and heating methods, while urban water and electricity are necessary conditions for residents' family life. Therefore, the correlation between urban gas supply and urban population is significantly weaker.

Urban gas mainly includes natural gas and liquefied petroleum gas. Table 2 shows the scaling parameter values of different types of urban gas.

According to Table 2, the urban natural gas supply scales super-linearly with urban population and urban GRP and the urban LPG supply scales linearly with urban population and urban GRP, but the household LPG consumption scales sub-linearly with urban population and urban GRP. Generally speaking, urban natural gas in cities is mainly transported by natural gas pipelines, while liquefied petroleum gas is mainly supplied by gas tanks. With the increase in city scale, the urban gas supply gradually shifts from liquefied petroleum gas, with

**Table 2. Regression results of urban gas supply concerning urban population and urban GRP.**

| LN(RV) | With respect to LN(POP): First, Second and Third tier cities (n = 114) | | | With respect to LN(GRP): First, Second and Third tier cities (n = 114) | | |
|---|---|---|---|---|---|---|
| | b | Linearity | Adj -R² | b | Linearity | Adj-R² |
| TGS | 1.247 | Super-L | 0.508 | 1.004 | L | 0.552 |
| TNGS | 1.439 | Super-L | 0.44 | 1.183 | Super-L | 0.488 |
| TNGS(FR) | 1.280 | Super-L | 0.403 | 1.025 | L | 0.423 |
| TGS-LPG | 1.046 | L | 0.27 | 0.917 | L | 0.335 |
| TGS-LPG(FR) | 0.837 | Sub-L | 0.196 | 0.713 | Sub-L | 0.227 |

**RV:** Response variable

**TGS:** Total gas supply; **TNGS:** Total natural gas supply; **TNGS(FR):** Total natural gas supply for residents; **TGS-LPG:** Total gas supply of LPG (Liquefied Petroleum Gas); **TGS-LPG(FR):** Total gas supply of LPG for residents

**Super-L:** Super-Linear; **L:** Linear; **Sub-L:** Sub-Linear; **Adj-R²**: Adjusted $R^2$.

lower safety and combustion efficiency, to natural gas, with higher safety and combustion efficiency. The urban residents also gradually abandon the use of liquefied petroleum gas and accept natural gas with unified transportation, so the super-linear scaling relationship between urban gas supply and urban population is mainly due to the increase in urban natural gas use.

Fig 2 shows that the increasing urban scale drives the construction of urban infrastructure, which leads to a considerable increase in the length of urban natural gas transmission pipelines, so the consumption of urban natural gas is greatly promoted. In conclusion, the super-linear scaling relationship between urban gas supply and urban population is mainly due to the rapid increase in the length of urban natural gas pipelines.

**3.1.2 Research on the scaling laws of different types of cities.** According to the classification results of 114 cities, the urban scaling laws of the 14 industrial cities, 27 commercial cities, 73 mixed-economy cities are explored separately.

Fig 3 shows the proportional amounts of the average values of urban development indicators in three different types compared with the total average.

As can be seen from Fig 3, all average values of development indicators in industrial cities are much lower than the total average, indicating that the third-tier-and-above industrial cities are relatively poorly developed. The number of commercial cities accounts for 23.7%, but all average values of development indicators in commercial cities are much lower than the total average, indicating that commercial cities are more attractive to Chinese people and have better development. The number of mixed-economy cities accounts for 64.0%, and the average values of their indicators are slightly under this value, but the gaps are smaller than industrial cities, indicating that Chinese mixed-economy cities are still in a period of development and transformation with no distinctive scaling characteristics.

Table 3 shows the scaling parameter calculation results of several urban development indicators of three different types of cities.

In Table 3, it can be seen that almost all urban development indicators fail to fit well with the urban population, and it seems that the development of China's industrial cities does not follow the general urban scaling laws. The rapid development of the secondary industry in industrial cities is worsening the ecological environment of cities, thus leading to the migration of urban residents, which can offset the immigration of urban population attracted by economic growth, so the population growth in industrial cities did not increase significantly with urban development. However, the development of industrial cities will lead to the increase of GRP inevitably, so the urban development indicators of industrial cities show good correlations with urban GRP.

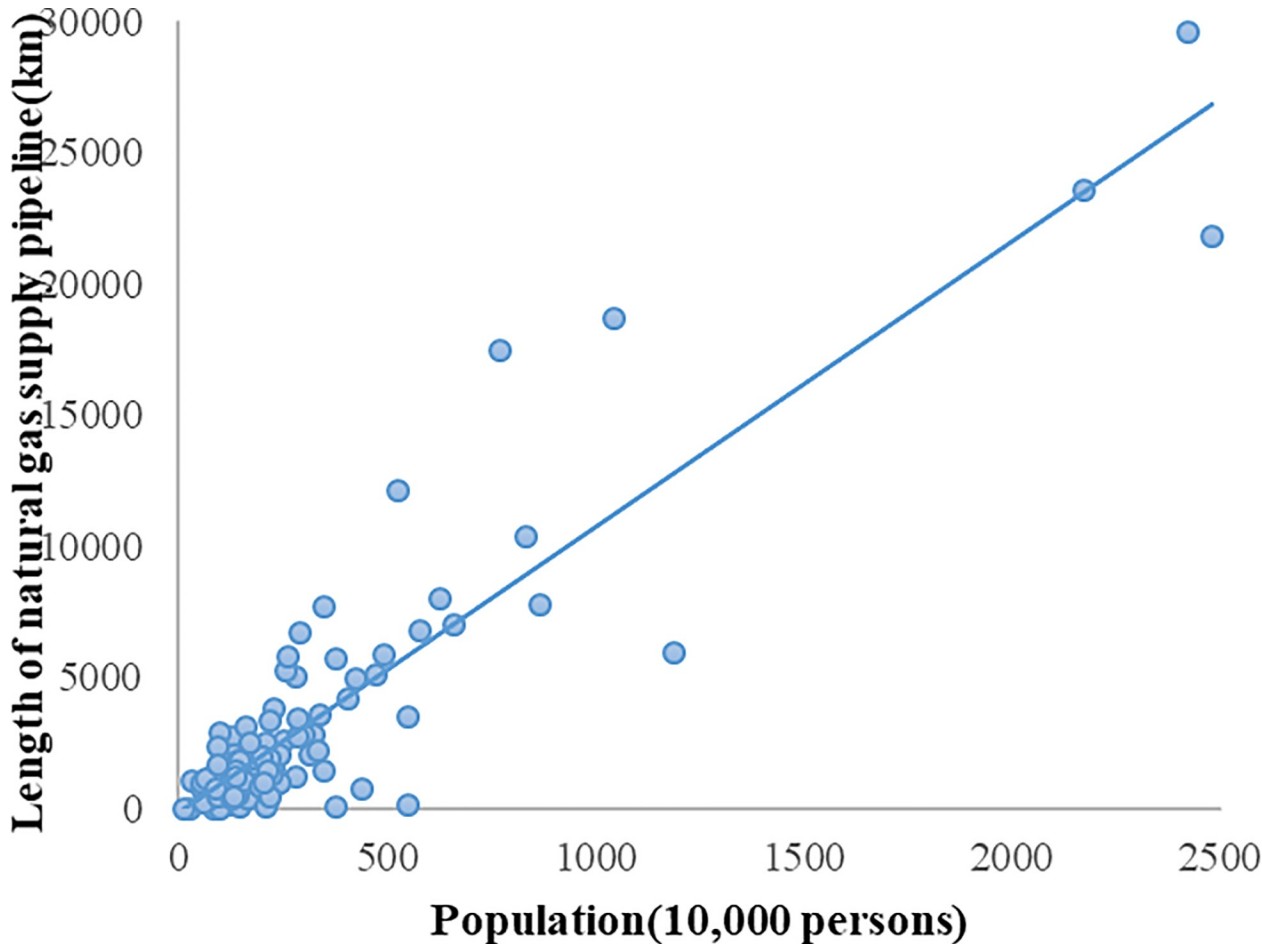

**Fig 2. Trend of urban natural gas pipeline length with the urban population.** (A) The blue bubble corresponds to the logarithm of the population and the length of the gas pipeline; (B) The blue oblique line represents the logarithmic regression line between urban population and urban natural gas pipeline length.

The urban scaling characteristics of commercial cities conform to the general urban scaling laws basically and their urban scaling laws are the most apparent. Urban development indicators show perfect fitting effects with urban population and GRP in commercial cities. Commercial cities mainly depend on the development of the tertiary industry. The production and consumption of goods and the existence of consumers are critical factors for commercial urban development. Therefore, for commercial cities, more urban population and more potential consumers will bring faster urban development and higher urban GRP. In China, the most developed cities are all commercial cities. On the whole, the development of Chinese commercial cities is relatively mature and their scaling laws are more visible.

The urban development of mixed-economy cities combines the development characteristics of the other two types of cities. Although the urban development indicators have discernable correlations with urban population and urban GRP, the correlation intensity is weaker than that of commercial cities and stronger than that of industrial cities. For mixed-economy cities, balancing the development of the secondary industry and tertiary industry is an important issue. The ambiguity of the urban type attribute will affect the formulation of urban policies, thus reducing the attractiveness and development potential of cities, which will ultimately affect the health of urban development.

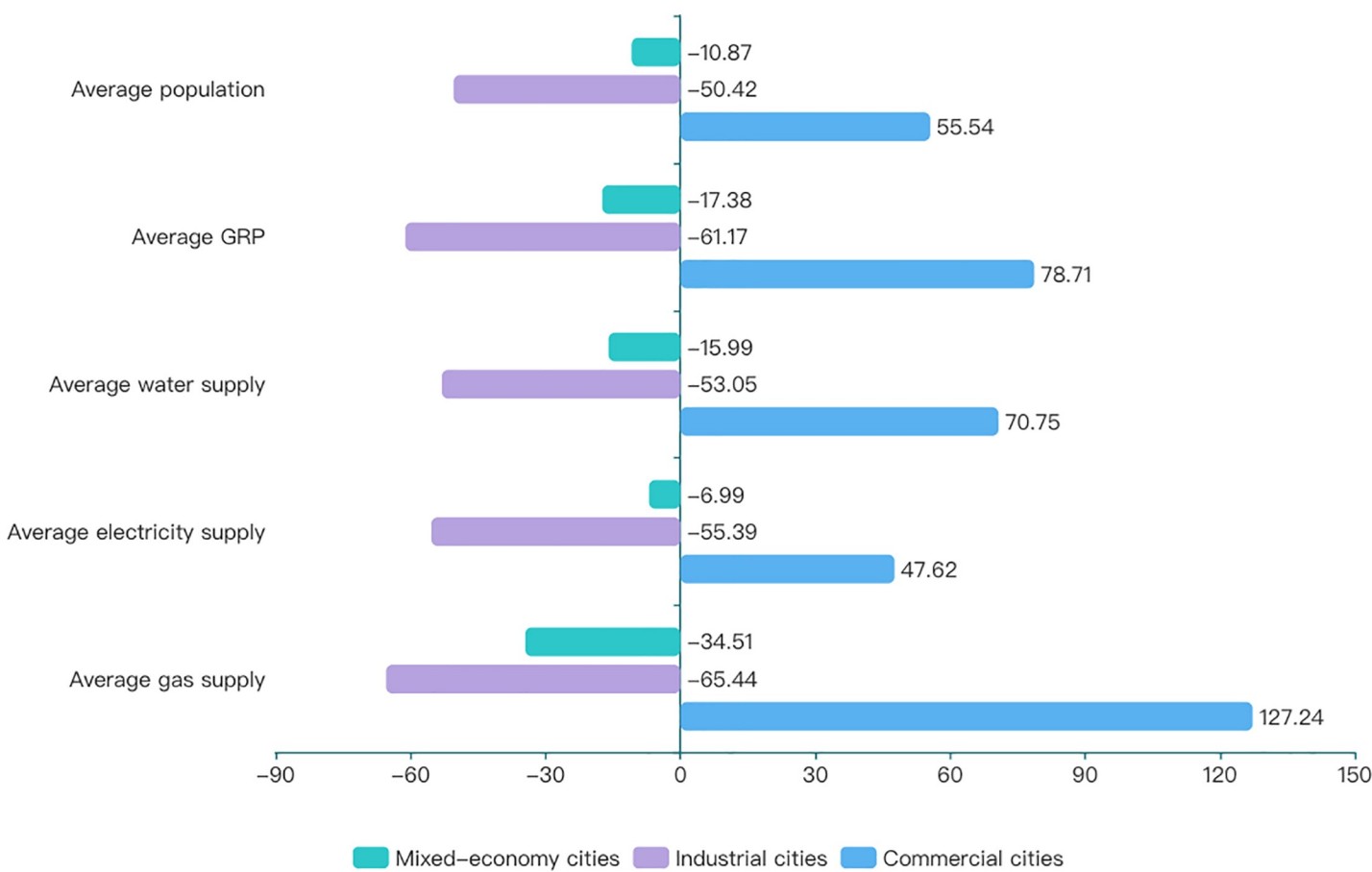

**Fig 3. Proportions of average values of development indicators for different types of cities compared with the total average.** (A) The dark blue bars, light blue bars and gray blue bars respectively represent the proportions of the average values of urban development indicators in mixed economy cities, industrial cities, commercial cities compared with the total average.

## 3.2 Research on influencing factors of urban electricity consumption

Among the indicators reflecting the state of urban development, urban electricity consumption is an essential one. Electricity is an indispensable resource in the process of industrial development and urban residents' living, and the massive consumption of electricity will inevitably exacerbate the destruction of the ecological environment. The univariate and multivariate regression analysis of influencing factors of urban electricity consumption for different types of cities can help us put forward some advice for efficient electricity use.

**3.2.1 Univariate regression analysis of factors affecting urban electricity consumption.** Table 4 lists the univariate regression results of the urban area, population, construction land area, street lamp number and per capita GRP concerning total electricity consumption, industrial electricity consumption, and household electricity consumption.

Correlation analysis of the urban electricity consumption reveals the population size effect, urban form effect and urban infrastructure effect, that is, urban population, construction land area and street lamp number positively correlate with urban electricity consumption. However, urban area and per capita GRP have little impact on urban electricity consumption. Urban area includes not only construction land but also land to be developed. The land area to be developed in different cities is very different from and consumes less electricity than construction land, so the urban area is weakly related to urban electricity consumption. The per

**Table 3. Scaling parameters of urban development indicators with urban population and GRP in different city types.**

| City type | LN(RV) | With respect to LN(Population) | | | With respect to LN(GRP) | | |
|---|---|---|---|---|---|---|---|
| | | b | Linearity | Adj-R² | b(Std.Error) | Linearity | Adj -R² |
| City-I (n = 14) | TGS | -1.236 | Negative | 0.063 | -0.368 | - | -0.046 |
| | TWS | -0.769 | - | -0.002 | 1.218 | Super-L | 0.502 |
| | WSEC | 0.924 | L | 0.029 | 1.487 | Super-L | 0.674 |
| | FAI | -0.287 | - | -0.072 | 0.062 | - | -0.082 |
| | CLA | 0.078 | - | -0.08 | 0.750 | Sub-L | 0.601 |
| | DPL | 0.432 | - | -0.06 | 0.982 | L | 0.269 |
| | RA | -0.374 | - | -0.031 | 0.751 | Sub-L | 0.523 |
| | PA | -0.127 | - | -0.079 | 0.380 | Sub-L | 0.025 |
| | GCA | -1.241 | Negative | 0.200 | 0.622 | Sub-L | 0.119 |
| | SLN | 0.536 | Sub-L | 0.031 | 0.326 | Sub-L | 0.038 |
| | LN(GDP)~LN(POP):b = 0.133, Adjusted R^2 = -0.076 | | | | | | |
| City-C (n = 27) | TGS | 1.125 | Super-L | 0.689 | 0.876 | L | 0.646 |
| | TWS | 1.067 | L | 0.875 | 0.880 | L | 0.901 |
| | WSEC | 1.067 | L | 0.927 | 0.853 | L | 0.9 |
| | FAI | 0.907 | L | 0.834 | 0.765 | Sub-L | 0.9 |
| | CLA | 0.873 | L | 0.875 | 0.725 | Sub-L | 0.902 |
| | DPL | 0.826 | Sub-L | 0.743 | 0.740 | Sub-L | 0.908 |
| | RA | 0.840 | Sub-L | 0.78 | 0.727 | Sub-L | 0.888 |
| | PA | 0.750 | Sub-L | 0.61 | 0.657 | Sub-L | 0.712 |
| | GCA | 0.985 | L | 0.756 | 0.880 | L | 0.921 |
| | SLN | 0.723 | Sub-L | 0.607 | 0.649 | Sub-L | 0.746 |
| | LN(GDP)~LN(POP):b = 1.148, Adjusted R^2 = 0.868 | | | | | | |
| City-M (n = 73) | TGS | 1.296 | Super-L | 0.458 | 1.067 | L | 0.543 |
| | TWS | 1.042 | L | 0.649 | 0.889 | L | 0.823 |
| | WSEC | 0.900 | L | 0.515 | 0.827 | Sub-L | 0.763 |
| | FAI | 1.024 | L | 0.633 | 0.883 | L | 0.823 |
| | CLA | 0.888 | L | 0.73 | 0.707 | Sub-L | 0.818 |
| | DPL | 0.988 | L | 0.621 | 0.802 | Sub-L | 0.714 |
| | RA | 0.912 | L | 0.638 | 0.743 | Sub-L | 0.739 |
| | PA | 0.802 | Sub-L | 0.446 | 0.657 | Sub-L | 0.523 |
| | GCA | 0.900 | L | 0.600 | 0.754 | Sub-L | 0.735 |
| | SLN | 0.810 | Sub-L | 0.509 | 0.721 | Sub-L | 0.705 |
| | LN(GDP)~LN(POP):b = 1.077, Adjusted R² = 0.663 | | | | | | |

City-I: Industrial cities; City-C: Commercial cities; City-M: Mixed-economy cities; RV: response variable; TGS: Total gas supply; TWS: Total water supply; WSEC: Whole society electricity consumption; FAI: Fixed asset investment; CLA: Construction land area; DPL: Drainage pipe length; RA: Road area; PA: Park area; GCA: Green coverage area; SLN: Street lamp number. Super-L: Super-Linear; L: Linear; Sub-L: Sub-Linear; Adj-R²:Adjusted R².

capita GRP can be used to measure the affluence of the residents in different regions, but the affluence of urban residents does not have much effect on consumption of essential living resources such as electricity, so the per capita GRP and electricity consumption present a weak correlation.

Table 5 lists the univariate regression results of total electricity consumption, industrial electricity consumption and household electricity consumption relative to the urban area, population, construction land area, street lamp number and per capita GRP in different types of cities.

**Table 4. Univariate regression results of electricity consumption in Chinese third-tier-and-above cities.**

| LN(IV) | With respect to LN(electricity use): First, second and third cities(n = 114) | | | | | | | | |
|---|---|---|---|---|---|---|---|---|---|
| | City-wide(WSEC) | | | Industry | | | Household | | |
| | b | Linearity | Adj-R² | b | Linearity | Adj-R² | b | Linearity | Adj-R² |
| UA | 0.473 | Sub-L | 0.153 | 0.459 | Sub-L | 0.104 | 0.487 | Sub-L | 0.18 |
| POP | 0.994 | L | 0.669 | 1.033 | L | 0.532 | 1.003 | L | 0.755 |
| CIA | 1.051 | L | 0.738 | 1.087 | L | 0.574 | 1.011 | L | 0.763 |
| SLN | 0.961 | L | 0.64 | 1.021 | L | 0.533 | 0.941 | L | 0.684 |
| P-GRP | 0.908 | L | 0.193 | 0.985 | L | 0.167 | 0.841 | Sub-L | 0.18 |

IV: Independent variable; WSEC: Whole society electricity consumption

UA: Urban area; Pop: Population; CLA: Construction land area; SLN: Street Lamp number; P-GRP: Per capita GRP

Super-L: Super-Linear; L: Linear; Sub-L: Sub-Linear; Adj-R²: Adjusted $R^2$.

The compositions of electricity consumption in different types of cities are different. According to Fig 4, it can be found that the power consumption of industrial cities is mainly concentrated in industrial electricity consumption, while household electricity consumption and other electricity consumption are relatively small. Compared with industrial cities, commercial cities and mixed-economy cities use significantly less industrial electricity, and household electricity and other types of electricity consume relatively more. The different compositions of electricity consumption may affect the scale characteristics of electricity consumption in different types of cities.

According to Table 5, the total electricity consumption of industrial cities has strong super-linear scaling relationships with urban construction land area and weak correlation with the

**Table 5. Univariate regression results of urban electricity consumption in different types of cities.**

| City type | LN(IV) | With respect to LN(EU) | | | | | | | | |
|---|---|---|---|---|---|---|---|---|---|---|
| | | City-wide EU(WSEC) | | | Industry | | | Household | | |
| | | b | Linearity | Adj-R² | b | Linearity | Adj-R² | b | Linearity | Adj-R² |
| City-I (n = 14) | UA | 0.447 | Sub-L | 0.114 | 0.517 | Sub-L | 0.097 | 0.114 | Sub-L | -0.068 |
| | POP | 0.924 | L | 0.019 | 0.883 | L | -0.019 | 1.532 | Super-L | 0.324 |
| | CLA | 1.706 | Super-L | 0.805 | 2.094 | Super-L | 0.828 | 0.494 | Sub-L | 0.049 |
| | SLN | 0.978 | L | 0.230 | 1.105 | Super-L | 0.189 | 0.822 | Sub-L | 0.281 |
| | P-GRP | 0.983 | L | 0.164 | 1.287 | Super-L | 0.206 | -0.096 | Negative | -0.087 |
| City-C (n = 27) | UA | 0.689 | Sub-L | 0.274 | 0.739 | Sub-L | 0.187 | 0.735 | Sub-L | 0.348 |
| | POP | 1.067 | L | 0.927 | 1.254 | Super-L | 0.800 | 1.010 | L | 0.903 |
| | CLA | 1.105 | Super-L | 0.86 | 1.264 | Super-L | 0.704 | 1.044 | L | 0.862 |
| | SLN | 1.106 | Super-L | 0.765 | 1.339 | Super-L | 0.701 | 1.038 | L | 0.730 |
| | P-GRP | 1.782 | Super-L | 0.364 | 2.050 | Super-L | 0.296 | 1.842 | Super-L | 0.430 |
| City-M (n = 73) | UA | 0.293 | Sub-L | 0.055 | 0.250 | Sub-L | 0.028 | 0.325 | Sub-L | 0.082 |
| | POP | 0.900 | L | 0.515 | 0.840 | Sub-L | 0.375 | 0.943 | L | 0.644 |
| | CLA | 0.990 | L | 0.668 | 0.959 | L | 0.525 | 0.981 | L | 0.737 |
| | SLN | 0.848 | Sub-L | 0.586 | 0.834 | Sub-L | 0.475 | 0.847 | Sub-L | 0.667 |
| | P-GRP | 0.662 | Sub-L | 0.146 | 0.690 | Sub-L | 0.132 | 0.616 | Sub-L | 0.142 |

City-I: Industrial cities; City-C: Commercial cities; City-M: Mixed-economy cities

IV: Independent variable; EU: Electricity use; WSEC: Whole society electricity consumption; UA: Urban area; POP: Population; CLA: Construction land area; SLN: Street lamp number; P-GRP: Per capita GRP

Super-L: Super-Linear; L: Linear; Sub-L: Sub-Linear; Adj-R²: Adjusted $R^2$.

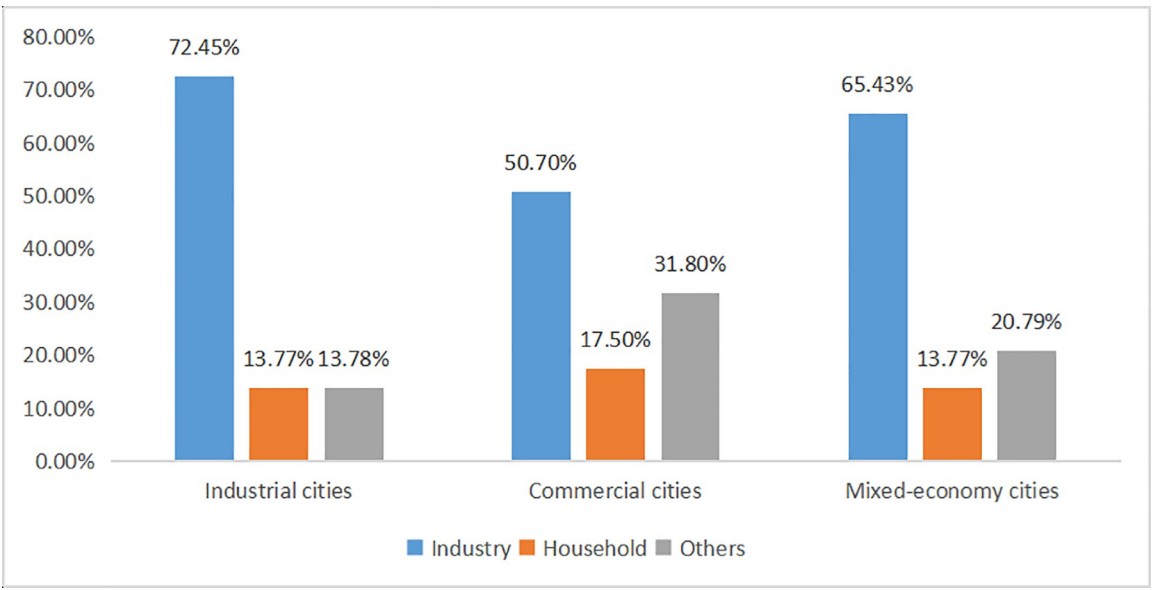

**Fig 4. Electricity consumption compositions of different types of cities.** (A) The blue bars, brown bars and gray bars represent the proportions of industrial electricity, commercial electricity and other types of electricity respectively.

urban population. The industrial electricity consumption of industrial cities accounts for a large proportion, so that the electricity consumption of industrial cities is mainly affected by industrial development. The development of industrial cities depends on the production of industrial enterprises, which are engaged in the exploitation and processing of natural resources. For industrial cities, the increase of construction land area usually means the development of urban industry, which requires further exploitation of resources. Natural resource exploitation requires a lot of electricity. As a result, the urban electricity consumption will rise super-linearly as construction land area increases. In the process of developing from a coal mining city to a comprehensive industrial city with petroleum, iron, steel and other industries, industrial cities will further aggravate urban electricity consumption.

The electricity consumption of commercial cities and mixed-economy cities reflects the impact of the urban population size effect, urban form effect and urban infrastructure effect, and is little affected by urban area and per capita GRP. The differences of the two types of cities are the $R^2$ values of regression models.

**3.2.2 Multivariate regression analysis of factors affecting urban electricity consumption.** Table 6 shows the multivariate regression results of factors influencing the electricity consumption in Chinese third-tier-and-above cities.

According to the multivariate regression results in Table 6, urban area and per capita GRP have little impact on the urban electricity consumption of commercial cities, and urban area shows a negative influence. The scaling of the urban area and the growth of urban per capita GRP reflect the rapid transformation of urban commercialization. A large number of labor-intensive urban enterprises may move their production plants to less-developed cities with abundant human resources and low labor costs. Besides, the optimization of urban electricity efficiency in more-developed cities may also alleviate the city's electricity burden. Therefore, the scaling of the urban area and the growth of per capita GRP could slow down the increase in urban electricity consumption for commercial cities.

Urban population, urban construction land area and street lamp number have significant positive influences on urban electricity consumption. The urban population and urban

**Table 6. Multivariate regression results of urban electricity consumption.**

| LN(EU) | City type | LN(Independent variable) | | | | | Adj- R2 |
|---|---|---|---|---|---|---|---|
| | | b | | | | | |
| | | Urban area | Population | Construction land area | Street lamp number | Per capita GRP | |
| City-wide EU (WSEC) | City-A | - | 0.322** | 0.769** | - | - | 0.753 |
| | City-I | - | 0.793* | 1.682*** | - | - | 0.870 |
| | City-C | -0.180* | 0.750*** | 0.471** | - | - | 0.952 |
| | City-M | - | - | 0.702*** | 0.323* | - | 0.694 |
| Industrial EU | City-A | - | 0.582*** | - | 0.578*** | - | 0.599 |
| | City-I | - | - | 2.094*** | - | - | 0.828 |
| | City-C | - | 1.254* | - | - | - | 0.800 |
| | City-M | - | - | 0.644*** | 0.352* | - | 0.549 |
| Household EU | City-A | - | 0.441*** | 0.371** | 0.293*** | - | 0.827 |
| | City-I | - | 1.282* | - | 0.669* | - | 0.515 |
| | City-C | - | 0.881*** | - | - | 0.626*** | 0.939 |
| | City-M | - | - | 0.657*** | 0.360*** | - | 0.776 |

**City-A:** All cities; **City-I:** Industrial cities; **City-C:** Commercial cities; **City-M:** Mixed-economy cities; **EU:** Electricity use; **WSEC:** Whole society electricity consumption; **Adj-R$^2$**: Adjusted R$^2$.

* Indicates whether the urban scaling parameter value is different from 0 (***p < 0.001; **p < 0.01; *p < 0.05;).

construction land area reflect the development trend of cities indirectly. The increase of urban population and construction land indicates the positive development of the urban economy and thus promotes the increase of urban electricity consumption. Street lamp number represents the city's urban infrastructure construction level. Urban infrastructure includes energy facilities, transportation facilities and communication facilities, all of which are pure consumers of electricity resources. More street lamps correlate with better infrastructure construction, thus street lamp numbers show positive relationships with urban electricity consumption.

According to the b values in the multivariate regression models, when urban area, population and construction land area increase by 10%, the total electricity consumption of commercial cities will decrease by 1.80%, increase by 7.50% and increase by 4.71% respectively. When population increases by 10%, the industrial electricity consumption of commercial cities will increase by 12.54%. When other parameters remain unchanged, the per capita GRP in cities will increase by 10%, and the household electricity consumption in commercial cities will increase by 6.26%.

Comparing the results of the electricity consumption regression analysis between Table 5 and Table 6, there are many differences in the values of urban scale factors affecting urban electricity consumption. This indicates that the influence factors of urban electricity consumption are sensitive to the inclusion of other variables.

On the whole, the results of multivariate regression analysis of the factors affecting urban electricity consumption are consistent with the results of univariate regression analysis. The regression results show that Chinese urban electricity consumption is mainly affected by urban population, urban construction land area, and street lamp number, while urban area and per capita GRP have little impact on electricity consumption.

## 4. Conclusion

In the context of 114 Chinese third-tier-and-above cities, our results show that the overall development patterns in China are consistent with the general urban scaling laws. Previous

studies have rarely been conducted within Chinese cities, and our research can help readers understand the scaling characteristics of them. Urban innovation wealth indicators scale super-linearly with urban population changes, urban infrastructure indicators scale sub-linearly with urban population changes, and urban material and energy indicators related to individual demands, except urban gas supply, scale linearly with urban population changes. The urban gas supply shows super-linear scaling with the urban population because of the rapidly increasing provision of urban natural gas transmission pipelines. In addition, the study analyzed the scaling of urban development indicators with GRP which has rarely used as independent variable. The goodness of fit with urban GRP as the independent variable appears better than that with the urban population, which manifests that the urban scaling characteristics of Chinese cities could be better modeled by urban GRP. The consistency between urban scaling laws and Chinese urban scaling characteristics makes it possible for China to promote urban development by referring to the experience of other countries. In the process of expanding the size of Chinese cities, in addition to considering the needs of the urban population for various development indicators, it should also be considered that the rapid development of the urban economy will also lead to increasing requirements for various indicators. Urban economy should play a more important role in the urban planning and construction process. In the context of all cities in China vigorously introducing talents, it should be considered whether the city's economy is sufficient to support the city's sustainable development. The development of Chinese cities should be based on people and more on economy.

For different types of cities, the differences between the values of scaling parameters indicate different development characteristics. The influence of city type on the characteristics of urban scaling has always been ignored. In industrial cities, the urban development indicators have no apparent correlation with urban population, but correlate strongly with urban GRP. Although the development of secondary industry in industrial cities can promote the development of urban GRP, it is challenging to attract talent. Also, the development of industry causes the deterioration of the ecological environment. While the Internet economy and service economy are taking up a greater and greater proportion of the national economy, the traditional industrial economy seems to show more weakness. How industrial cities can balance the relationships between economic development and ecological environment will be a question needing careful thought. For mixed-economy cities, their unclear urban attribute makes their development behave in a less straightforward way. The mixed-economy cities need to refer to the development experience of other types of cities for further development. The Chinese government should think about the differences between different types of cities and adjust measures to local conditions when making decisions on urban development. Industrial cities can properly transfer economic development centers to commercial development, and at the same time need to coordinate the relationship between the ecological environment and industrial development. Commercial cities need to pay more attention to building livable cities and attract more residents. Mixed-economy cities need clarify the center and direction of development, so as to improve the speed and quality of development.

Univariate and multivariate analyses of factors influencing electricity consumption show that urban electricity consumption is mainly affected by urban population, urban construction land area and street lamp number. To reduce the cities' electricity consumption, urban residents should pay more attention to saving electricity, and more power-saving facilities should be adopted under urban infrastructure construction. In addition, urban area and per capita GRP have little impact on electricity consumption. Although the correlations are weak, increasing the per capita GRP of urban residents can still play a significant role in slowing down the increase of urban electricity consumption. To fundamentally solve the environmental pollution problems of urban electricity use, cities need to rely on new technologies to

improve the efficiency of urban electricity and develop cleaner energy sources such as solar energy and wind energy.

In the future, further analysis on the development characteristics of China's industrial cities could help build urban scaling models with more generality and utility. Furthermore, the impact of urban development on the biophysical environment is also worthy of further investigation.

## Supporting information

**S1 Table. Detailed information of various indicators.**
(DOC)

## Author Contributions

**Data curation:** Xingchao Liu.

**Methodology:** Xingchao Liu.

**Supervision:** Zhihong Zou.

**Writing – original draft:** Xingchao Liu.

**Writing – review & editing:** Xingchao Liu, Zhihong Zou.

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
