## [Decision Letter · Decision Letter 0]

20 Apr 2020

PONE-D-20-08745

Analysis on urban scaling characteristics of China’s relatively developed cities

PLOS ONE

Dear Mr. Xingchao,

Thank you for submitting your manuscript to PLOS ONE. After careful consideration, we feel that it has merit but does not fully meet PLOS ONE’s publication criteria as it currently stands. Therefore, we invite you to submit a revised version of the manuscript that addresses the points raised during the review process.

We would appreciate receiving your revised manuscript by Jun 04 2020 11:59PM. To enhance the reproducibility of your results, we recommend that if applicable you deposit your laboratory protocols in protocols.io, where a protocol can be assigned its own identifier (DOI) such that it can be cited independently in the future. For instructions see: http://journals.plos.org/plosone/s/submission-guidelines#loc-laboratory-protocols

We look forward to receiving your revised manuscript.

Kind regards,

Bing Xue, Ph.D.

Academic Editor

PLOS ONE

Journal Requirements:

1. We suggest you thoroughly copyedit your manuscript for language usage, spelling, and grammar. If you do not know anyone who can help you do this, you may wish to consider employing a professional scientific editing service.  

3. Thank you for including the following funding information within the acknoweldgements section of your manuscript; "The research was supported by the National Natural Science Foundation of China (No. 51478025)"

Reviewers' comments:

Reviewer's Responses to Questions

**Comments to the Author**

1. Is the manuscript technically sound, and do the data support the conclusions?

Reviewer #1: Yes

Reviewer #2: Partly

Reviewer #3: Yes

Reviewer #4: Yes

2. Has the statistical analysis been performed appropriately and rigorously? 

Reviewer #1: Yes

Reviewer #2: Yes

Reviewer #3: I Don't Know

Reviewer #4: Yes

3. Have the authors made all data underlying the findings in their manuscript fully available?

Reviewer #1: Yes

Reviewer #2: Yes

Reviewer #3: No

Reviewer #4: Yes

4. Is the manuscript presented in an intelligible fashion and written in standard English?

Reviewer #1: Yes

Reviewer #2: Yes

Reviewer #3: Yes

Reviewer #4: No

5. Review Comments to the Author

Reviewer #1: General comments:

1. 114 cities are not a decent sample size, especially when the sample is further classified into three groups. At such small sample size, especially the 14 industrial cities, how do you make sure the regression results are robust?

2. why use supply instead of gas and water consumption data while the electricity consumption data is used? Are there factors such as import of gas, water resource allocation would affect supply and consumption?

3. A table showing more details of the data used in the study would help the readers to know better about your research.

4. I don’t think park area is a good example for urban infrastructure, especially when explaining its sub-linear relationship with population or GRP.

5. The results in Table 5 should be explained with more details, for example the difference between industry consumption and household consumption. Different city types may have different constitution of industry and household consumption and whether it causes the super-linear scaling in industrial cities, but not commercial and mixed cities.

6. if you can access the electricity consumption data by different economic sector in each city, that would be an interesting perspective to see the scaling.

Specific comments:

line 180: how the 263 cities were selected, what are the relationship between 114 third-tier-and-above cities, 338 prefecture-level cities and 263 cities?

line 205: why compare with 263 cities? rather than 338 cities or 114 cities.

line 213: does the yearbook you mentioned provides all data at municipal districts level?

line 291: the sentence should be re-written to clarify which variables are individual needs, for example the drainage pipeline is individual needs or urban infrastructure?

line 336: “while urban water and water”

Reviewer #2: Based on urban scaling quantitative models, this paper analyzes the correlation among various index variables in urban and population and GRP, as well as the factors affecting urban electriurban consumption, taking 114 third-tier-and-above Chinese urban as examples. The amount of research data is large, the content is substantial, and the research conclusion has certain practical significance, but there are still the following problems.

(1)It is suggested that the introduction should be reorganized. On the one hand, the content of urban crime mentioned in the introduction is not highly related to the article. On the other hand, the introduction discusses the size and scale of the urban, but at the end of the article, it is mentioned that the analysis is based on the type of urban, and the relationship between the two is not high.

(2)In this paper, the urban scaling characteristics are mentioned and the third-tier-and-above urban are selected for analysis. However, the analysis process of the full text is mostly based on the urban type, and there is less research on the urban scale. It is suggested to add comparative analysis of the characteristics of different scale of urban in the analysis, and the research value may be better.

(3)The percentages of various indicators in different types of cities are compared and analyzed in Fig. 3, but it can be seen that there are great differences in the number of commercial cities, industrial cities and mixed-economy cities. Comparing the percentages of various cities with the percentages of consumed resources directly, the reliability and reliability of the results are not high. For comparative analysis, it is recommended to compare the average values of various cities.

(4)The article uses a lot of space to analyze the influencing factors of urban electricity consumption, which is not closely related to the theme of this article. Electricity consumption is an important indicator of urban scaling. The indicators ,such as the scale of urban construction land, urban population, water consumption, and transportation, can be used as key indicators that affect the size of the city. What are the reasons and particularities of analyzing the power consumption of the city?

(5)The conclusions mentioned in this paper can be used to formulate effective targeted policies for cities and to provide suggestions for reducing resource consumption in the process of urbanization. However, the end of this paper describes this part too little, and it is suggested to elaborate on different types or cities of different scales.

It is recommended to review after modification.

Reviewer #3: This manuscript analyse the urban scaling characteristics of China’s cites, the topic is interesting. However the novelty of this manuscript is lack. I did not see new findings on the urban scaling. The resluts of in figure 1 and 2 are not fresh knowledges.

For the indicators selection. the street lamp number was eclected as urban infrastructure indicator. In my opinions, the building and food supply are more important infrastructure than street lamp.

figure 2 shows the results of relationship of natural gas supply pipe line with population, but in some provinces in China, there have no such natural gas pipe due to lack of this kind of resource.

Reviewer #4: Urban scaling is an important research direction in geography and economics. Within the scope of 114 third-tierand-above Chinese cities, this article calculate the scaling parameters of various urban development variables with respect to urban population and urban GRP in different city types based on urban scaling quantitative models.

1. Introduction is long enough, and it is suggested that a part of literature review can be added. such as "1. Introduction; 2.Literature review;…"

2.Research data needs descriptive statistical analysis of the data and more detailed explanation of the data source and processing.

3.According to "2016 China Business Charm Ranking", there are 119 third-tier-and-above cities, Why this article selected 114 third-tier-and-above cities ?

4.What are the 25 variables related to urban scaling and why are they chosen?

5.If the research results can be compared with the existing research, the research contribution of this article will be enhanced.

6. PLOS authors have the option to publish the peer review history of their article (what does this mean?). If published, this will include your full peer review and any attached files.

Reviewer #1: Yes: Long Chen

Reviewer #2: No

Reviewer #3: No

Reviewer #4: No

---

## [Author Response · Author response to Decision Letter 0]

3 Jun 2020

Dear Editors and Reviewers:

Thank you for your letter and for the reviewers’ comments concerning our manuscript entitled “Analysis on urban scaling characteristics of China’s relatively developed cities” (ID: PONE-D-20-08745). Those comments are all valuable and very helpful for revising and improving our paper, as well as the important guiding significance to our researches. We have studied comments carefully and have made correction which we hope meet with approval. Revised portions are marked in the paper.

The main corrections in the paper and the responds to the reviewer’s comments are as following.

Responds to the reviewer’s comments:

Reviewer #1: 

General comments:

1. 114 cities are not a decent sample size, especially when the sample is further classified into three groups. At such small sample size, especially the 14 industrial cities, how do you make sure the regression results are robust? 

Response:

The number of cities is not an important factor affecting the regression results. It can be found from Table 2 that the number of commercial cities is only 27, and the number of mixed-economy cities is 73. However, the regression effect of commercial cities is significantly better than that of mixed economy cities, which to some extent indicates that the number of different types of cities will not affect the reliability of the results. In fact, we have also calculated the scaling index of different urban development indicators with urban population and urban GRP under more industrial cities (including the fourth-tier and below industrial cities, a total of 42), and the results are generally consistent with the results of this study, so I think that even if the number of industrial cities is relatively small, it will not have a big impact on the reliability of the regression results. It is the great difference between different industrial cities that leads to the less obvious scaling characteristics of industrial cities.

2. Why use supply instead of gas and water consumption data while the electricity consumption data is used? Are there factors such as import of gas, water resource allocation would affect supply and consumption?

Response:

The total amount of urban gas supply refers to the total amount of gas supplied by gas enterprises to users during the reporting period, including the amount of sales and losses. Total urban water supply refers to the total amount of water supplied by water supply units during the reporting period, including effective water supply and leakage water. Leakage of gas and water mainly refers to the leakage caused by damage to pipelines and ancillary facilities, theft or failure of counting tables in the process of water supply. Leakage can help us judge whether the construction of urban infrastructure has been improved to some extent with the expansion of city scale. Therefore, the total water supply and gas supply were used in our study, rather than the consumption index after the leakage was removed. Urban power loss is usually caused by electricity transportation, which is an inevitable quantitative technical loss. Therefore, the city statistical yearbook usually only collects electricity consumption data and does not include electricity loss data. Therefore, we also use electricity consumption data instead of electricity supply data in our research.

3. A table showing more details of the data used in the study would help the readers to know better about your research.

Response:

According to your suggestion, we have made a table to help readers understand the specific information of the research indicators, including the indicator name, unit, specific classification and data source. This table has been included in the supporting information.

4. I don’t think park area is a good example for urban infrastructure, especially when explaining its sub-linear relationship with population or GRP.

Response:

Urban park is an important part of China's urban construction planning, and the park area has always been an important indicator to measure the level of China's urban modernization. The park plays an important role in helping to improve the city's ecological environment and residents' living environment. Therefore, we believe that park area should be an important part of urban infrastructure.

Of course, we carefully considered your opinion and decided to increase the "green area coverage" as one of the urban infrastructure indicators. The urban green space coverage area covers a wider range. In addition to the park green area, it also includes the city's residential green space and traffic green space, so it may be a better indicator of urban infrastructure. The two indicators of comprehensive park area and green area coverage.

5. The results in Table 5 should be explained with more details, for example the difference between industry consumption and household consumption. Different city types may have different constitution of industry and household consumption and whether it causes the super-linear scaling in industrial cities, but not commercial and mixed cities.

Response:

Based on your suggestions, we calculated the urban electricity consumption proportions in different types of cities and analyzed its possible impact on the scale of urban electricity consumption. The modifications are shown in the article.

6. If you can access the electricity consumption data by different economic sector in each city, that would be an interesting perspective to see the scaling. 

Response:

We have worked hard to search the electricity consumption data of different economic sectors in various cities. This is indeed a very difficult thing because many data are missing. I'm sorry that this part of the content may not be supplemented in this research, but your suggestion is indeed very enlightening. It is a topic that is worth writing an article for in-depth research. We will try to fill this gap in the future. 

Specific comments: 

line 180: how the 263 cities were selected, what are the relationship between 114 third-tier-and-above cities, 338 prefecture-level cities and 263 cities?

Response:

China’s business charm rankings have divided 338 prefecture-level cities in China. We hope to conduct research within these cities. However, the China City Statistical Yearbook and China City Construction Statistical Yearbook do not include all 338 prefecture-level cities. At the same time, there are many data missing in the two yearbooks. After the cross-comparison and supplement of the data, only the data of 263 cities is relatively complete, so the data table containing 263 cities is used as the original data. 263 cities include all prefecture-level cities of different levels, and we only want to select more developed cities of third-tier and above to conduct research, so only 114 cities of third-tier and above are taken as the final research scope.

line 205: why compare with 263 cities? Rather than 338 cities or 114 cities.

Response:

The city type division method used in the study was conducted by comparing the proportions of different industries in each city with the national average. The 263 urban development data obtained are defaulted to represent the national average. So our compare is within the scope of the 263 cities, but not 338 or 114 cities.

line 213: does the yearbook you mentioned provides all data at municipal districts level?

Response:

Yes, these two yearbooks can provide all the data at the municipal level, among which some data are missing. We have filled in the data by consulting the provincial and municipal yearbooks and online databases.

line 291: the sentence should be re-written to clarify which variables are individual needs, for example the drainage pipeline is individual needs or urban infrastructure?

Response:

The table including the indicator name, unit, specific classification and data source has been included in the supporting information.

line 336: “while urban water and water”

Response:

I'm sorry, it was my fault. It should be “while urban water and electricity”. It has been modified. Sorry again.

Reviewer #2: 

1. It is suggested that the introduction should be reorganized. On the one hand, the content of urban crime mentioned in the introduction is not highly related to the article. On the other hand, the introduction discusses the size and scale of the urban, but at the end of the article, it is mentioned that the analysis is based on the type of urban, and the relationship between the two is not high.

Response:

Based on your suggestions, we have made the necessary cuts and modifications to the introduction.

1. The part of the city crime not discussed in this article has been deleted.

2. The impact of city size on city development is an important part of our research. Using the urban population to represent the size of the city is an important prerequisite for the Bettencourt scale model. On this basis, we have increased the urban GRP to represent the economic scale of the city. Therefore, the scaling indexes of various urban development indicators with the urban population and urban GRP reflects the impact of urban size on urban development. City type is an important dimension of our research. Through separate analysis of different types of cities, it can help us understand the differences in scalig characteristics between different types of cities. Therefore, city size and city type are two dimensions of research, rather than two variables with low correlation.

2. In this paper, the urban scaling characteristics are mentioned and the third-tier-and-above urban are selected for analysis. However, the analysis process of the full text is mostly based on the urban type, and there is less research on the urban scale. It is suggested to add comparative analysis of the characteristics of different scale of urban in the analysis, and the research value may be better.

Response:

The answer to this question is similar to the answer to the previous question. The impact of city size on urban development is an important part of our research. Using the urban population to represent the size of the city is an important prerequisite for the Bettencourt scale model. On this basis, we have added urban GRP to represent the urban economic scale. Therefore, the scaling indexes of various urban development indicators along with urban population and urban GRP reflects the impact of urban size on urban development. City type is an important dimension of our research. Through separate analysis of different types of cities, it can help us understand the differences of scaling characteristics between different types of cities.

3. The percentages of various indicators in different types of cities are compared and analyzed in Fig. 3, but it can be seen that there are great differences in the number of commercial cities, industrial cities and mixed-economy cities. Comparing the percentages of various cities with the percentages of consumed resources directly, the reliability and reliability of the results are not high. For comparative analysis, it is recommended to compare the average values of various cities.

Response:

Based on your suggestions, we have modified this part of the content and compared the average of different types of cities with the overall average, which is indeed more intuitive.

4. The article uses a lot of space to analyze the influencing factors of urban electricity consumption, which is not closely related to the theme of this article. Electricity consumption is an important indicator of urban scaling. The indicators ,such as the scale of urban construction land, urban population, water consumption, and transportation, can be used as key indicators that affect the size of the city. What are the reasons and particularities of analyzing the power consumption of the city?

Response:

Urban electricity is not only a necessary energy for the development of urban commerce and industry, but also an important energy for the living of urban residents. More importantly, urban electricity consumption is also an important source of urban CO2 emissions. Therefore, electric energy consumption is closely related to the ecological environment. At the same time, compared with other indicators, the urban statistical yearbook provides a more detailed dimension of urban energy consumption data, which provides the possibility for in-depth analysis of urban energy consumption. In-depth analysis of urban electricity consumption can not only provide a reference for energy conservation, but also help relieve the pressure on the ecological environment.

5. The conclusions mentioned in this paper can be used to formulate effective targeted policies for cities and to provide suggestions for reducing resource consumption in the process of urbanization. However, the end of this paper describes this part too little, and it is suggested to elaborate on different types or cities of different scales.

Response:

According to your suggestion, we have supplemented the conclusion. For different types of cities, we have put forward different policy recommendations.

Reviewer #3: 

1.This manuscript analyse the urban scaling characteristics of China’s cites, the topic is interesting. However the novelty of this manuscript is lack. I did not see new findings on the urban scaling. The resluts of in figure 1 and 2 are not fresh knowledges.

Response:

The law of urban expansion scale is the universal law of urban expansion established by Bettencourt based on the urban population, and it shows regularity without being restricted by space and time. China is rarely studied because of its complicated urbanization. We select more developed Chinese cities to try to verify whether the scaling law established by Bettencourt is applicable in China, and introduce urban GRP as an important independent variable. Although such research is not new, this universality provides a basis for China to master the urbanization process and make reference to the urbanization of other countries.

2.For the indicators selection. the street lamp number was eclected as urban infrastructure indicator. In my opinions, the building and food supply are more important infrastructure than street lamp.

Response:

The number of city street lights is closely related to the urban population and living environment, and is the most dense urban infrastructure. Urban street lights are often concentrated in areas with a high population density. The increase in the total number of street lights can reflect the population aggregation and infrastructure construction caused by urbanization. Therefore, the total number of urban street lights has been adopted by us as one of the infrastructure indicators. The city's architecture and food supply are relatively complex, which may involve many types of indicators. It may not be appropriate to conduct research directly as infrastructure, and it needs to be analyzed separately in subsequent studies.

3.Figure 2 shows the results of relationship of natural gas supply pipe line with population, but in some provinces in China, there have no such natural gas pipe due to lack of this kind of resource。

Response:

The data used in our research are all municipal districts. The natural gas pipeline facilities in more developed cities are relatively complete. China has also adopted the "west-to-east natural gas transmission" to complete the construction of pipeline facilities nationwide. According to our research results, it can be found that the urban natural gas pipeline shows a super-linear scaling relationship with the urban population, indicating that the construction of natural gas pipeline varies greatly among different cities, and that there is still much room for improvement in the construction of natural gas pipeline in Chinese cities.

Reviewer #4:

1. Introduction is long enough, and it is suggested that a part of literature review can be added. such as "1. Introduction; 2.Literature review;…"

Response:

We have cut and modified the introduction part of the article, but did not split it into two parts. In the introductory part, we hope to be able to tell the reader the background, research content, and significance of the article as if telling a story. Splitting the introductory part may cause the part to lose its integrity and be unclear. If you think there is still a problem with the revised introduction, we will modify it again according to your suggestions

2.Research data needs descriptive statistical analysis of the data and more detailed explanation of the data source and processing.

Response:

A table including the indicator name, unit, specific classification and data source has been included in the supporting information.

3.According to "2016 China Business Charm Ranking", there are 119 third-tier-and-above cities, Why this article selected 114 third-tier-and-above cities ?

Response:

According to the “2016 China Business Charm Ranking”, there are 119 third-tier cities and above in China, but the data of five of these cities (including three second-tier cities and two third-tier cities) are seriously missing or not included in China city statistical yearbook, so our study only adopt 114 third-tier cities and above.

4.What are the 25 variables related to urban scaling and why are they chosen?

Response:

The 25 variables associated with urban growth include: 

population density, per capita road area, urban area, urban population, built-up area, urban construction land area, public facility land, fixed asset investment, street lights number, road area, park area, land area, GRP, per capita GRP, total water supply, electricity consumption for the whole society, industrial electricity, residential electricity, total gas supply, number of buses, operating vehicles, number of taxis, green area coverage and Three industries account for the proportion of GRP.

The 25 variables are all from China's urban statistical yearbook and China's urban construction statistical yearbook, covering most of the indicators of urban innovation wealth, material energy or infrastructure, which are also important factors to measure the level of urbanization. Therefore, we selected these indicators for analysis and calculation.

5.If the research results can be compared with the existing research, the research contribution of this article will be enhanced.

Response:

Thank you for your suggestions and we have supplemented the conclusion. Comparing our research results with previous studies can help readers understand the significance of the research. 

We have tried our best to improve the manuscript and made some changes in the

manuscript. These changes will not influence the content and framework of the

paper. And here we did not list the changes but marked in revised paper.

We appreciate for editors and reviewers’ warm work earnestly and hope that the

correction will meet with approval.

Once again, thank you very much for your comments and suggestion.

Best regards,

Xingchao Liu and Zhihong Zou

---

## [Decision Letter · Decision Letter 1]

30 Jun 2020

PONE-D-20-08745R1

Analysis on urban scaling characteristics of China’s relatively developed cities

PLOS ONE

Dear Dr. Xingchao,

Thank you for submitting your manuscript to PLOS ONE. After careful consideration, we feel that it has merit but does not fully meet PLOS ONE’s publication criteria as it currently stands. Therefore, we invite you to submit a revised version of the manuscript that addresses the points raised during the review process.

We look forward to receiving your revised manuscript.

Kind regards,

Bing Xue, Ph.D.

Academic Editor

PLOS ONE

Reviewers' comments:

Reviewer's Responses to Questions

**Comments to the Author**

1. If the authors have adequately addressed your comments raised in a previous round of review and you feel that this manuscript is now acceptable for publication, you may indicate that here to bypass the “Comments to the Author” section, enter your conflict of interest statement in the “Confidential to Editor” section, and submit your "Accept" recommendation.

Reviewer #1: All comments have been addressed

Reviewer #2: All comments have been addressed

Reviewer #4: All comments have been addressed

2. Is the manuscript technically sound, and do the data support the conclusions?

Reviewer #1: Partly

Reviewer #2: Yes

Reviewer #4: Yes

3. Has the statistical analysis been performed appropriately and rigorously? 

Reviewer #1: No

Reviewer #2: Yes

Reviewer #4: Yes

4. Have the authors made all data underlying the findings in their manuscript fully available?

Reviewer #1: Yes

Reviewer #2: Yes

Reviewer #4: Yes

5. Is the manuscript presented in an intelligible fashion and written in standard English?

Reviewer #1: Yes

Reviewer #2: Yes

Reviewer #4: Yes

6. Review Comments to the Author

Reviewer #1: The authors have addressed most of my comments in the first round of review, I appreciate the thorough revisions by the authors. Many parts of the manuscript were greately improved in this round. However, I don't think one of my concerns, which is the small sample size for the 14 industrial cities has been addressed properly. My concern is from the pespective of statistics, saying that you need a decent sample size to conduct any regression analysis and to deliver valid results. Apparently 14 is not a decent size for regressions. The negative Adj-R2 in Table 2 also indicates that you have small R-square and a small sample size. One possible solution maybe the inclusion of panel data for the cities to increase the sample size. Beyond that, I have no further comments.

Reviewer #2: The data source of this article is true, the research method is feasible, and the research conclusion is credible. The author responds to the expert's opinions one by one and provides a detailed explanation.

Reviewer #4: The authors s have adequately addressed my comments, and I feel that this manuscript is now acceptable for publication

7. PLOS authors have the option to publish the peer review history of their article (what does this mean?). If published, this will include your full peer review and any attached files.

Reviewer #1: No

Reviewer #2: No

Reviewer #4: No

---

## [Author Response · Author response to Decision Letter 1]

7 Jul 2020

Dear Editors and Reviewers:

Thank you for your letter and for the reviewers’ comments concerning our manuscript entitled “Analysis on urban scaling characteristics of China’s relatively developed cities” (ID: PONE-D-20-08745). Those comments are all valuable and very helpful for revising and improving our paper, as well as the important guiding significance to our researches. We have studied comments carefully and have made correction which we hope meet with approval. 

The responds to the reviewer’s comments are as following.

Responds to the reviewer’s comments:

Reviewer #1:  

I don't think one of my concerns, which is the small sample size for the 14 industrial cities has been addressed properly. My concern is from the perspective of statistics, saying that you need a decent sample size to conduct any regression analysis and to deliver valid results. Apparently 14 is not a decent size for regressions. The negative Adj-R2 in Table 2 also indicates that you have small R-square and a small sample size. One possible solution maybe the inclusion of panel data for the cities to increase the sample size. 

Response to Reviewer #1:

According to your suggestion that we should increase the sample number of industrial cities, we selected 42 industrial cities for the calculation of the scaling index of urban development indicators with the change of urban population and urban GRP. The 42 industrial cities include not only 14 third-tier and above industrial cities, but also 28 industrial cities from the fourth and fifth tier.

Table 1 shows the scaling index results of the development indicators of 42 industrial cities with the change of urban population and urban GRP, and Table 2 shows the scaling index results of 14 third-tier and above industrial cities with the change of urban population and urban GRP.

From Table 1 and Table 2, it can be found that, even if the selection scope of industrial cities is expanded and the sample number of industrial cities is increased, the Adj-R2 obtained by 42 industrial cities is not significantly improved compared with the previous 14 industrial cities. In the regression results with urban population, Adj-R2 is mostly within the range of 0.1-0.3, which is still not a reliable R2 value. In the regression results with urban GRP, except that the R2 value of fixed asset investment has been significantly improved, the other R2 values are still at a low range level.

In fact, we can't find any more industrial cities in the third-tier and above.If we want to increase the sample size, we may need to include some underdeveloped industrial cities. However, this conflicts with our research topic, and does not result in more reliable regression results (R2 values did not increase significantly in 42 industrial cities).

The number of industrial cities in China is relatively small, and the development level and urbanization level of industrial cities are relatively low. The unremarkable scaling characteristics of industrial cities in the research results is more due to the lag of the development models of industrial cities. The development of industrial cities is more dependent on industry. Compared with commercial cities and mixed economy cities with a larger commercial proportion, they are less attractive to residents. The ecological environment deterioration caused by industrial development further intensifies the outflow of local residents. Therefore, urban physical energy indicators, infrastructure indicators and other development indicators that are closely related to residents have failed to perform well in proportion to the urban population. Therefore, the number of third-tier and above industrial cities in the study is not an important factor affecting the scaling results, but the development characteristics of industrial cities themselves are the main reason.

The development characteristics of China's industrial cities are quite different from those of other types of cities, suggesting that the Chinese government should take more measures in accordance with local conditions when making urban development policies.

In our study, industrial cities show very different development characteristics, which are often neglected in previous studies.It is worth further in-depth study to explore the development characteristics of industrial cities. We also hope to explore and analyze the scaling laws of industrial cities separately in the future, which is really an interesting and worthwhile subject.

We appreciate for editors and reviewers’ warm work earnestly and hope that the response will answer the reviewer’s question.

Once again, thank you very much for your comments and suggestion.

Best regards,

Xingchao Liu and Zhihong Zou

---

## [Editor Report · Decision Letter 2]

10 Jul 2020

Analysis on urban scaling characteristics of China’s relatively developed cities

PONE-D-20-08745R2

Dear Dr. Xingchao,

We’re pleased to inform you that your manuscript has been judged scientifically suitable for publication and will be formally accepted for publication once it meets all outstanding technical requirements.

Kind regards,

Bing Xue, Ph.D.

Academic Editor

PLOS ONE
---

## [Editor Report · Acceptance letter]

15 Jul 2020

PONE-D-20-08745R2 

Analysis on urban scaling characteristics of China’s relatively developed cities 

Dear Dr. Xingchao:

I'm pleased to inform you that your manuscript has been deemed suitable for publication in PLOS ONE. Congratulations! Your manuscript is now with our production department. 

Kind regards, 

on behalf of

Professor Bing Xue 

Academic Editor

PLOS ONE